# Redox-Sensitive Delivery of Doxorubicin from Nanoparticles of Poly(ethylene glycol)-Chitosan Copolymer for Treatment of Drug-Resistant Oral Cancer Cells

**DOI:** 10.3390/ijms241813704

**Published:** 2023-09-05

**Authors:** Kaengwon Yoon, Seunggon Jung, Jaeyoung Ryu, Hong-Ju Park, Hee-Kyun Oh, Min-Suk Kook

**Affiliations:** 1El-Dental Clinic, Seomun Daero Street 625, Namgu, Gwangju 61737, Republic of Korea; kaengwonyoon@outlook.kr; 2Department of Maxillofacial Oral Surgery, School of Dentistry, Chonnam National University, Gwangju 61186, Republic of Korea; seunggon.jung@jnu.ac.kr (S.J.); ryu@jnu.ac.kr (J.R.); omspark@jnu.ac.kr (H.-J.P.); hkoh@jnu.ac.kr (H.-K.O.)

**Keywords:** ROS-sensitive drug delivery, nanoparticles, drug-resistant, doxorubicin, oral squamous carcinoma cells

## Abstract

Reactive oxygen species (ROS)-sensitive polymer nanoparticles were synthesized for tumor targeting of an anticancer drug, doxorubicin (DOX). For this purpose, chitosan-methoxy poly(ethylene glycol) (mPEG) (ChitoPEG)-graft copolymer was synthesized and then DOX was conjugated to the backbone of chitosan using a thioketal linker. Subsequently, the chemical structure of the DOX-conjugated ChitoPEG copolymer (ChitoPEGthDOX) was confirmed via ^1^H nuclear magnetic resonance (NMR) spectra. Nanoparticles of the ChitoPEGthDOX conjugates have spherical shapes and a size of approximately 100 nm. Transmission electron microscopy (TEM) has shown that ChitoPEGthDOX nanoparticles disintegrate in the presence of hydrogen peroxide and the particle size distribution also changes from a monomodal/narrow distribution pattern to a multi-modal/wide distribution pattern. Furthermore, DOX is released faster in the presence of hydrogen peroxide. These results indicated that ChitoPEGthDOX nanoparticles have ROS sensitivity. The anticancer activity of the nanoparticles was evaluated using AT84 oral squamous carcinoma cells. Moreover, DOX-resistant AT84 cells were prepared in vitro. DOX and its nanoparticles showed dose-dependent cytotoxicity in both DOX-sensitive and DOX-resistant AT84 cells in vitro. However, DOX itself showed reduced cytotoxicity against DOX-resistant AT84 cells, while the nanoparticles showed almost similar cytotoxicity to DOX-sensitive and DOX-resistant AT84 cells. This result may be due to the inhibition of intracellular delivery of free DOX, while nanoparticles were efficiently internalized in DOX-resistant cells. The in vivo study of a DOX-resistant AT84 cell-bearing tumor xenograft model showed that nanoparticles have higher antitumor efficacy than those found in free DOX treatment. These results may be related to the efficient accumulation of nanoparticles in the tumor tissue, i.e., the fluorescence intensity in the tumor tissue was stronger than that of any other organs. Our findings suggest that ChitoPEGthDOX nanoparticles may be a promising candidate for ROS-sensitive anticancer delivery against DOX-resistant oral cancer cells.

## 1. Introduction

The physiological status of the tumor microenvironment is normally quite different from the healthy microenvironment. Abnormal biochemical properties, such as acidic pH, high oxidative stress, multi-drug resistance, enhanced metabolism, and overexpression of molecular receptors, are common [1,2,3,4,5]. These abnormal properties of the tumor microenvironment impede the delivery of traditional anticancer drugs to the tumor tissue, minimize the therapeutic efficacy, and cause undesirable side-effects against normal tissues. During chemotherapy, an acidic pH or elevated redox potential of the tumor tissues can disturb the delivery potential of anticancer drugs into tumor tissues and frequently induce drug resistance [6,7]. In particular, when oxidative stress is elevated in the tumor microenvironment, reactive oxygen species (ROS) increase, which is significantly associated with cancer progression, i.e., ROS are known to promote proliferation, migration, and stemness of the cancer cells in the tumor microenvironment [8].

The incidence and mortality of oral cancer, one of the most threatening malignancies among all cancers, is continuously increasing worldwide [9]. Although oral cancer can be easily detected at an early stage, it is frequently diagnosed at an advanced stage [9,10]. Various regimens have been employed to treat oral cancer, but chemotherapy remains the preferred treatment regimen [10,11,12,13]. Adverse effects of chemotherapy, such as toxic side effects and oral mucositis, are still problematic to patients [11,12,13,14,15]. Moreover, the low specificity of the chemotherapeutic agents and the drug resistance of traditional anticancer agents against oral cancer have also contributed to a high recurrence rate and low survival among those affected [16,17]. Elevated oxidative stress is significantly related to drug resistance and cancer progression, i.e., as a chemotherapeutic agent, such as doxorubicin increases intracellular ROS, antioxidant capacity also increases to maintain cellular homeostasis, leading to high levels of drug-resistant cancer cells [18,19]. Olivier et al. reported that a high level of ROS in glioblastoma cells plays an essential role in drug resistance and then these cellular events are frequently quoted to inhibit the use of chemotherapy [20]. Hence, a ROS-mediated delivery system for anticancer agents is required to overcome these problems.

Stimuli-responsive nanocarriers, such as nanoparticles, polymeric micelles, and polymeric conjugates, have been spotlighted over two decades because they have the potential to deliver anticancer agents into tumor tissues by overcoming various tumor barriers [21,22,23,24,25]. For example, the abnormal physiological status of the tumor microenvironment provides an opportunity for drug targeting using nanocarriers, i.e., nanoparticles can be designed to be sensitive to an acidic pH and liberate anticancer agents in the acidic tumor microenvironment [24]. Elevated redox potential of the tumor microenvironment can also be used to address targeting issues using nanoparticles [25,26]. When anticancer agents are conjugated to nanoparticles via ROS or GSH-sensitive linkers, nanoparticles are able to liberate anticancer agents at a higher ROS or GSH level. These properties facilitate the delivery of anticancer agents that are sensitive to these conditions [25,26]. For example, Lim et al. reported that pH-sensitive nanoparticles can improve tumor-specific delivery of a radiosensitizer in a mouse brain tumor model and efficiently provide radiation-induced anticancer therapy against the brain tumor [23]. Lee and Jeong reported that poly(L-histidine) (PHS)-conjugated hyaluronic acid (HA) nanoparticles with disulfide linkages are sensitive to an acidic pH and an elevated level of GSH in the tumor microenvironment [24]. They argued that the drug release rate from PHS-conjugated HA nanoparticles is accelerated under an acidic pH and high GSH levels, which are intrinsic properties of tumor physiology. Furthermore, Deng et al. reported that ROS-inducible and pH-sensitive nanocarriers can improve the delivery of doxorubicin (DOX) under tumor hypoxia, and thus overcome drug resistance [25]. When ROS-sensitive nanoparticles with phenyl-boronic acid pinacol esters and diselenide linkages sensitively react with hydrogen peroxide, the release of the anticancer drug can be accelerated at a higher ROS level [26]. They argued that the anticancer activity was higher at higher levels of ROS in vitro. The physiology of tumor microenvironments is normally specified as a higher ROS level. This status stimulates the release of the anticancer agent from nanoparticles through the process of ROS-specific degradation, while nanoparticles maintain stability in the bloodstream.

In this study, we synthesized doxorubicin (DOX)-conjugated chitosan-g-poly(ethylene glycol) (PEG) (ChitoPEG) copolymer nanoparticles with thioketal linkages (ChitoPEGthDOX) for the stimuli-sensitive delivery of DOX against oral cancer cells. Because thioketal linkages are degradable in the oxidative stress, nanoparticles of ChitoPEGthDOX can collapse as a result of oxidative stress and then the DOX release rate can be accelerated. We characterized the ChitoPEGthDOX nanoparticles, assessed the action against oral cancer cells in vitro, and evaluated the anticancer activity in vivo using an animal tumor xenograft model.

## 2. Results and Discussion

### 2.1. Synthesis and Characterization of ChitoPEGthDOX Conjugates and Nanoparticles

Non-ionic and hydrophilic mPEG was introduced into the chitosan backbone to synthesize a ChitoPEG graft copolymer, as reported previously [27] (Figure 1a). ^1^H NMR-spectra showed that ethylene protons were confirmed at 3.4~3.6 ppm (Figure 1b) and H2~H6 of the chitosan backbone was confirmed at 3.0~3.8 ppm. The H1 proton of chitosan was confirmed at 4.2~4.5 ppm. Based on the weight measurement, the degree of substitution of PEG against the chitosan backbone was approximately 1 PEG unit/10.2 glucosamine unit.

Specific peaks of ThdCOOH (Figure 2a,b), and DOX (Figure 2c,d) were confirmed at 1.0~4.0 ppm and 1.0~8.0 ppm, respectively. One end carboxylic acid group of the ROS-sensitive linkage ThdCOOH was attached to the amine group of DOX, and then another carboxylic acid group was activated using the EDAC/NHS system as shown in Figure 3a. This was then attached to the amine group of the chitosan backbone. As shown in Figure 3b, specific peaks of the ChitoPEG copolymer, ThdCOOH, and DOX were confirmed at 1.0~4.6 ppm. Since PEG has hydrophilic and stealth properties in blood circulation, ChitoPEGthDOX conjugates may form core–shell nanoparticles, i.e., PEG forms the outer-shell of nanoparticles, while the DOX-thioketal linker-chitosan part forms the core of the nanoparticles [26,28]. Since DOX has hydrophobic properties, DOX aggregated in the core of the nanoparticles. The DOX contents in the ChitoPEGthDOX conjugates were estimated using a UV absorption measurement, as shown in Table 1. The DOX content was approximately 8.7% (*w*/*w*), while the theoretical value was 9.6% (*w*/*w*). These results may be due to DOX being lost during the synthesis procedure. Nanoparticles of ChitoPEGthDOX conjugates have a small particle size of approximately 100 nm (Table 1).

ChitoPEGthDOX nanoparticles in the aqueous solution were spherical (Figure 4A(a)). The diameter of the nanoparticles was approximately 100 nm. The particle size distribution of the nanoparticles was monomodal and narrow (Figure 4B(a)). These results were quite similar to the results in Table 1. Since the thioketal linker has ROS-sensitive properties and can be degraded by oxidative stress [29], hydrogen peroxide was added to the aqueous solution of ChitoPEGthDOX nanoparticles. As shown in Figure 4A(b,c),B(b,c), the morphology of the nanoparticles was significantly changed, i.e., the nanoparticles were disintegrated by hydrogen peroxide, resulting in particle sizes that were multimodal and wide. These results indicated that biologically, ChitoPEGthDOX nanoparticles can be disintegrated under oxidative stress. Yoon et al. also reported that nanoparticles with thioketal linkers can be disintegrated by hydrogen peroxide, and thus facilitate the liberation of anticancer agents [29]. Similarly, we showed that the DOX release rate of ChitoPEGthDOX nanoparticles was accelerated in the presence of hydrogen peroxide, and the acceleration rate was determined by the hydrogen peroxide concentration (Figure 5). Furthermore, our results also indicated that ChitoPEGthDOX nanoparticles are responsive to ROS, and thus facilitate the liberation of anticancer agents in the tumor tissues in the presence of oxidative stress.

### 2.2. In Vitro Anticancer Activity against AT84 OSCC Cells

The anticancer activity of the DOX or ChitoPEGthDOX nanoparticles was assessed in vitro using AT84 OSCC cells. To induce DOX-resistance cells, the cells were exposed to a non-toxic concentration of DOX for 3 h. Subsequently, the DOX concentration was gradually increased to 0.1 µg/mL. During this protocol, AT84 cells became resistant to DOX treatment (Figure 6), i.e., the viability of DOX-sensitive cells was higher than 80% up to 0.5 μg/mL of DOX, while more than 80% of DOX-resistant cells survived until 1.5 μg/mL of DOX (Figure 6a,b). When the nanoparticles of ChitoPEGthDOX were treated, cell viability in DOX-resistant cells was not significantly changed compared to that in DOX-sensitive cells. The ChitoPEG copolymer itself showed little toxicity in these concentration ranges for both DOX-sensitive and DOX-resistant cells, i.e., the cell viability was higher than 80% until 100 μg/mL of DOX in DOX-sensitive and DOX-resistant cells. Furthermore, the IC_50_ value of free DOX was significantly higher in DOX-resistant cells than in DOX-sensitive cells, while the nanoparticle value showed relatively few changes (Table 2). These results indicated that the ChitoPEGthDOX nanoparticles can overcome drug resistance in cancer cells. Figure 7 supported these results, i.e., the red fluorescence intensity of the free DOX treatment was not significantly increased (from 30 min to 120 min) in DOX-resistant cells while the nanoparticles revealed strong fluorescence intensity, and the intensity gradually increased over time. As shown in Figure 8, the flow cytometric analysis also supported these results, i.e., nanoparticle treatment significantly increased the fluorescence intensity. These results indicated that ChitoPEGthDOX nanoparticles can overcome drug resistance and efficiently deliver anticancer drugs intracellularly.

### 2.3. In Vivo Study Using Tumor-Bearing Mice

The particle size is known to govern the fate of the nanoparticles biologically [30,31,32]. He et al. reported that the particle size of nanoparticles governs in vivo fate after administration [31]. The authors argued that nanoparticles with a particle size of 150 nm were efficiently accumulated in tumor tissues. Notably, Caster et al. reported the importance of the particle size of nanoparticles in chemoradiotherapy, i.e., nanoparticles with diameters of 100 nm resulted in the smallest tumor growth rate compared to nanoparticles of 50 nm or 150 nm diameters [32]. Furthermore, the authors argued that the smallest nanoparticles do not promise the best anticancer activity even though they are effective when it comes to avoiding hepatic and splenic accumulations [32]. Our nanoparticles have a small diameter of approximately 100 nm (Figure 4 and Table 1), which is a suitable particle size for site-specific targeting of tumor tissues and for promising anticancer activity. For in vivo study, DOX-resistant AT84 cells were implanted into the back of mice and the antitumor activity of ChitoPEGthDOX nanoparticles was evaluated (Figure 9). As shown in Figure 9a, the tumor volume gradually increased in the control treatment. DOX treatment inhibited tumor growth more significantly compared to control treatment. Importantly, tumor growth was efficiently inhibited by ChitoPEGthDOX nanoparticles, and the tumor volume was smallest among all the treatment groups, i.e., the tumor volume of the ChitoPEGthDOX nanoparticle treatment was less than 50% compared to the free DOX treatment. These results indicated that ChitoPEGthDOX nanoparticles have superior antitumor activity compared to free DOX. Figure 9b supported these results, i.e., the fluorescence intensity was significantly stronger in tumor tissues. Furthermore, the strongest fluorescence intensity was observed in the tumor tissues of all organs. These results indicated that ChitoPEGthDOX nanoparticles can efficiently target tumor tissues, thus inhibiting the growth of the tumor.

## 3. Materials and Methods

### 3.1. Chemicals

Chitosan (molecular weight (M.W.): 7000 g/mol, deacetylation degree > 97%, water- soluble) was purchased from Kittolife Co., Ltd. (Seoul, Republic of Korea). Doxorubicin hydrochloride (DOX) was obtained from LC Labs. Woburn, MA, USA). Thioketal dicarboxylic acid (Th-dCOOH) was purchased from RuixiBiotech Co., Ltd. (Xi’an, China). N-(3-dimethylaminopropyl)-N’-ethylcarbodiimide hydrochloride (EDAC), N-hydroxy succinimide (NHS), dimethyl sulfoxide (DMSO), 2′,7′-dichlorofluorescin diacetate (DCFH-DA), 3-(4,5-dimethyl2-thiazolyl)-2,5-diphenyl-2H-tetrazolium bromide (MTT), triethylamine (TEA) and 2,2,2-tribromoethanol (avertin) were purchased from Sigma Aldrich Chem. Co. (St. Louis, MO, USA). Dialysis membranes with molecular weight cutoffs (MWCO) of 1000 Da, 8000 Da, and 12,000 Da were purchased from Spectrum Labs., Inc. (Rancho Dominguez, CA, USA). Methoxy poly(ethylene glycol) N-hydroxysuccinimide (mPEG-NHS) (M.W. = 5000 g/mol) was purchased from Sunbio Co. Inc. (Seoul, Republic of Korea). Cell culture media and related materials were purchased from Invitrogen (New York, NY, USA). All other chemicals and organic solvents were used as extra-pure grade.

### 3.2. Synthesis DOX-Conjugated ChitoPEG Having Thioketal Linker (ChitoPEGthDOX)

ChitoPEG copolymer: ChitoPEG copolymer was synthesized as reported previously [27]. Chitosan (180 mg) was dissolved in DMSO/H_2_O (7/3 (*v*/*v*), 10 mL), and then mPEG-NHS (500 mg) was added to this solution. This solution was reacted for 2 days. Then, it was dialyzed against water for 2 days using a dialysis membrane (molecular weight cut-off size (MWCO) = 12,000 g/mol) to remove unreacted reactants and byproducts. Water was exchanged every 3~4 h intervals and the resulting solution was lyophilized for 3 days. The lyophilized solids were precipitated into an excess amount of chloroform to remove the unreacted mPEG. Then, the precipitants were dried under a vacuum to obtain a yellowish solid. This was refrigerated until used.

ChitoPEGthDOX conjugates: DOX HCl (21.5 mg, 37.1 μM) was dissolved in 3 mL of DMSO and one drop of TEA was added. An equivalent mole of ThdCOOH (8.83 mg) was dissolved in DMSO with EDAC (7.2 mg) and NHS (4.3 mg) to activate one end carboxylic acid of the ThdCOOH. This solution was stirred for 6 h. The ThdCOOH solution was added to the DOX HCl solution and then further stirred for 12 h to conjugate DOX with the ThdCOOH (DOXth conjugates). ChitoPEG copolymers (193 mg) in DMSO/H_2_O (7/3 (*v*/*v*), 10 mL) were mixed with the DOXth conjugates solution. This solution was stirred for 24 h and, after that, introduced into a dialysis membrane (MWCO = 8000 g/mol) and dialyzed against deionized water for 1 day, with water exchanges at 3 h intervals. Following this, the resulting solution was lyophilized for 3 days to obtain the ChitoPEGthDOX conjugates.

### 3.3. Fabrication and Characterization of Nanoparticles of ChitoPEGthDOX

ChitoPEGthDOX conjugates (20 mg) were reconstituted in deionized water (3 mL), and 5 mL DMSO was added. This was dialyzed against water for 24 h using a dialysis membrane (MWCO = 8000 g/mol) to remove the organic solvent. The resulting solution was then used for analysis or experiment.

^1^H nuclear magnetic resonance (NMR) spectroscopy (500 mHz superconducting Fourier transform (FT)-NMR spectrometer, Varian Unity Inova 500 MHz NB High-Resolution FT NMR; Varian Inc., Santa Clara, CA, USA) was employed to confirm the synthesis procedure. Synthesized chemicals were dissolved in D_2_O, DMSO, or D_2_O/DMSO mixtures.

The morphology of the nanoparticles was observed with a transmission electron microscope (TEM) (H-7600, Hitachi Instruments Ltd., Tokyo, Japan). An aqueous solution of ChitoPEGthDOX (nanoparticle weight 1 mg/mL) was dropped onto a carbon film-coated copper grid. This was negatively stained with phosphotungstic acid (0.1%, *w*/*w* in H_2_O). The observation was carried out at 80 kV and 20 °C.

An aqueous nanoparticle solution was used to analyze the particle size distribution using Nano-ZS (Malvern, Worcestershire, UK) at 20 °C. The nanoparticle concentration was less than 0.1 wt-%.

### 3.4. Drug Release Study

An aqueous solution of nanoparticles fabricated as described above was adjusted to 20 mL using deionized water (1 mg nanoparticles/mL). This solution (5 mL) was introduced into the dialysis membrane (MWCO = 8000 g/mol) and then put into a 50 mL conical tube with 45 mL of phosphate-buffered saline (PBS, pH 7.4, 0.01 M). The in vitro drug release study was performed at 100 rpm and 37 °C using the shaking incubator. At predetermined time intervals, PBS in the conical tube was retrieved and the level of the released drug was measured. Subsequently, the PBS was replaced. The released DOX in the harvested PBS solution was used to measure at 489 nm using an ultraviolet spectrophotometer (UV-1601, Shimadzu Co., Ltd., Osaka, Japan). To correct the absorption of DOX concentration, a similar concentration of empty ChitoPEG conjugates was used. All experiments were performed in triplicate, and the results were expressed as mean ± S.D.

### 3.5. Cell Culture Study

The murine oral squamous cell carcinoma AT84 cells were maintained with RPMI1640 media supplemented with 10% FBS and 1% antibiotics. AT84 cells were provided by Dr E.J. Shillitoe (State University of New York, Upstate Medical University) as described previously [32]. For the preparation of DOX-resistant AT84 cells, AT84 cells were exposed to DOX HCl (0.001 µg/mL) for 3h, replaced with fresh media, and then cultured in an CO_2_ incubator (5% CO_2_, 37 °C) for 2 days. The intermittent exposure of DOX HCl against cancer cells was repeated 3 times. The DOX HCl concentration in these procedures was gradually increased to 0.1 µg/mL.

Cell cytotoxicity: AT84 cells (1 × 10^4^ cells) were seeded in 96 wells, and then cultured in the CO_2_ incubator (5% CO_2_, 37 °C) overnight. After that, the media were discarded and replaced with fresh media containing DOX HCl, ChitoPEGthDOX nanoparticles, and ChitoPEG copolymer. For the DOX HCl treatment, DOX HCl dissolved in DMSO was diluted with serum-free RPMI1640 media until 0.5% *v*/*v*. ChitoPEGthDOX nanoparticles, and ChitoPEG copolymer in aqueous solution, were sterilized with a 0.8 µm syringe filter (Whatman Inc., Maidstone, England) and then diluted with a serum-free media. The DOX concentration in the sterilized sample was measured again using UV spectrophotometer and corrected. Cells were cultured in 5% CO_2_ and incubated at 37 °C. One or two days later, MTT solution (30 µL, 5 mg MTT/ mL in PBS) was added to the cell culture and then cultured in 5% CO_2_ and incubated at 37 °C. After 3 h, the supernatants were discarded and replaced with DMSO (100 µL). The cell viability was measured at 570 nm using a microplate reader (Infinite M200 pro multimode microplate readers, Tecan Trading AG Inc., Männedorf, Switzerland). The cell viability was calculated from 8 wells and expressed as mean ± SD.

To observe cell morphology, DOX-resistant AT84 (1 × 10^5^ cells/well) was seeded onto the cover glass in 6 wells. The cells were treated with DOX HCl or ChitoPEGthDOX nanoparticles for 0.5~2 h and then washed with PBS. After that, the cells were fixed with 10% paraformaldehyde solution for 10 min, immobilized with immobilization solution (Immunomount, Thermo Electron Co. Pittsburgh, PA, USA) for 1 h and then observed under a fluorescence microscope (Eclipse 80i; Nikon, Tokyo, Japan).

Flow cytometry analysis was performed as follows: DOX-resistant AT84 cells (1 × 10^6^ cells) were treated with free DOX or nanoparticles (5 µg/mL DOX concentration) for 0.5~2 h. After that, the cells were washed with PBS twice, and then harvested via centrifugation. Flow cytometry analysis was performed using Invitrogen Attune NxT flow cytometers (ThermoFisher Scientific, Waltham, MA, USA).

### 3.6. Tumor Xenograft Using Animals for In Vivo Anticancer Activity Study

For evaluation of in vivo anticancer activity of the nanoparticles, a tumor-xenograft model of male nude BALb/C mice was used (20 g, 4 weeks old). AT84 cells (1 × 10^6^) were subcutaneously (s.c.) administered into the backs of mice, and the mice were freely fed food/water until further treatment. When the size of the solid tumor became 4~5 mm in diameter, mice were divided into three groups: control, DOX HCl treatment, and nanoparticle treatment. Three days after tumor cell implantation, DOX or nanoparticles were administered intravenously (i.v.). For control treatment, PBS (100 µL) was administered i.v. For DOX HCl treatment, DOX HCl was dissolved in PBS and then sterilized with a 0.8 µm syringe filter. For nanoparticle treatment, the ChitoPEGthDOX nanoparticles were also sterilized with a 0.8 µm syringe filter. The DOX concentration of the DOX HCl treatment and the nanoparticle treatment was corrected via absorption measurement using a UV spectrophotometer. The dose of DOX in the DOX HCl treatment and the nanoparticle treatment was both 5 mg/kg (Injection volume: 100 μL). After 25 days, all mice were sacrificed, and tumor tissues were harvested to compare tumor weight. The changes in tumor volume were measured with vernier calipers at 5-day intervals. The following equation was used to calculate the tumor volume: tumor volume (mm^3^) = (length × width^2^)/2.

For in vivo imaging of tumor-bearing mice, ChitoPEGthDOX nanoparticles in PBS were injected i.v. via the tail vein. Whole-body imaging of the mice was observed with an animal imaging instrument (MaestroTM^2^, Cambridge Research & Instrumentation, Inc., Hopkinton, MA, USA). Fluorescence images of the mice were observed after the animals were sacrificed.

The results were analyzed statistically using the *t*-test, with *p* < 0.01 as the minimal level of significance.

## 4. Conclusions

In this study, we synthesized ChitoPEGthDOX conjugates for the ROS-sensitive delivery of anticancer agents against AT84 cells. DOX was conjugated with the chitosan backbone of the ChitoPEG copolymer using the thioketal linker. The thioketal linker was introduced between the chitosan backbone and DOX for ROS-sensitive degradation. TEM images and particle size measurement showed that ChitoPEGthDOX nanoparticles were sensitive to the presence of hydrogen peroxide, and disintegrated under the tested hydrogen peroxide concentration. In the drug release study, the DOX release rate was accelerated according to the hydrogen peroxide concentrations, indicating that ChitoPEGthDOX nanoparticles can be specifically disintegrated by ROS, and then *facilitate the release of anticancer agents under the oxidative stress*. DOX-resistant AT84 cells were prepared in vitro. In the cell viability study, the viability of DOX-sensitive AT84 cells gradually decreased according to the concentration of DOX. However, DOX-resistant AT84 cells became resistant to free DOX treatment and more than 80% of the cells survived until 1.5 μg/mL DOX concentration. Interestingly, ChitoPEGthDOX nanoparticles showed similar cell viability tendencies in both DOX-sensitive cells and DOX-resistant cells. Fluorescence observation and flow cytometric analysis supported these results, i.e., the intracellular delivery of ChitoPEGthDOX nanoparticles was higher than that of free DOX. The in vivo study using a tumor xenograft model of DOX-resistant AT84 cells demonstrated that the antitumor activity of ChitoPEGthDOX nanoparticles was significantly higher than that achieved during the free DOX treatment. Fluorescence observation of the mice supported these results, i.e., the fluorescence intensity in the tumor tissue was stronger than that in other organs. These results indicated that ChitoPEGthDOX nanoparticles were efficiently accumulated in the tumor tissue. Our findings suggest that ChitoPEGthDOX nanoparticles are a promising candidate for ROS-sensitive anticancer delivery against DOX-resistant oral cancer cells.

## Figures and Tables

**Figure 1 ijms-24-13704-f001:**
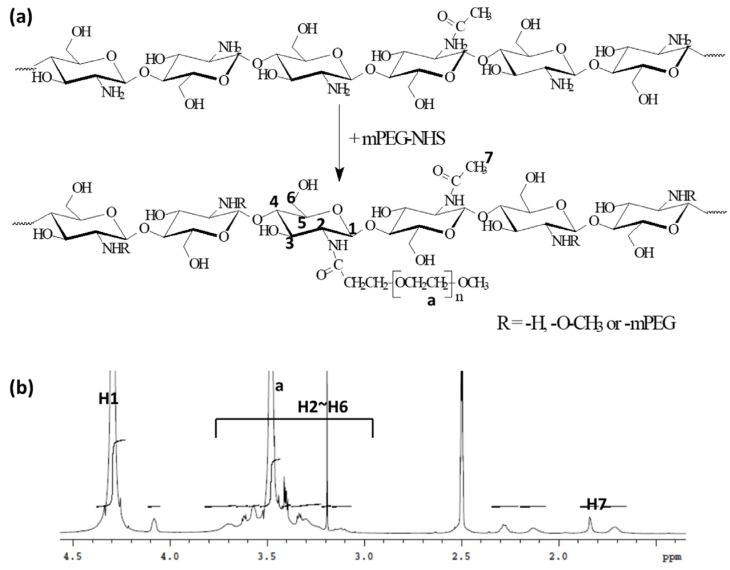
The synthesis scheme (**a**) and ^1^H NMR spectra (**b**) of ChitoPEG copolymer. The N-hydroxysuccinimide (NHS)-activated mPEG (mPEG-NHS) was conjugated with the amine groups of the chitosan backbone to form the ChitoPEG graft copolymer.

**Figure 2 ijms-24-13704-f002:**
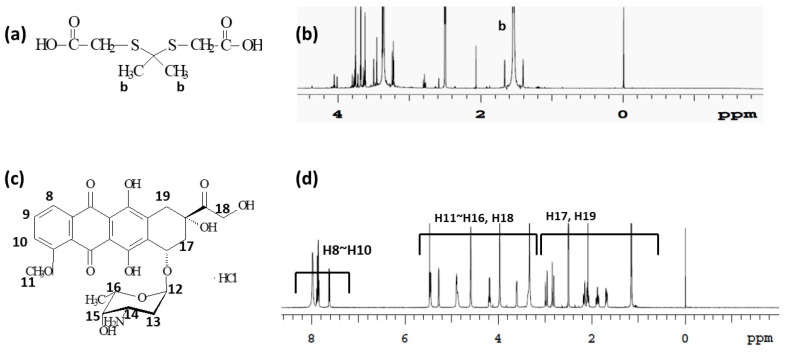
Chemical structure (**a**) and the ^1^H NMR spectra (**b**) of ThdCOOH. Chemical structure (**c**) and the ^1^H NMR spectra (**d**) of DOX HCl.

**Figure 3 ijms-24-13704-f003:**
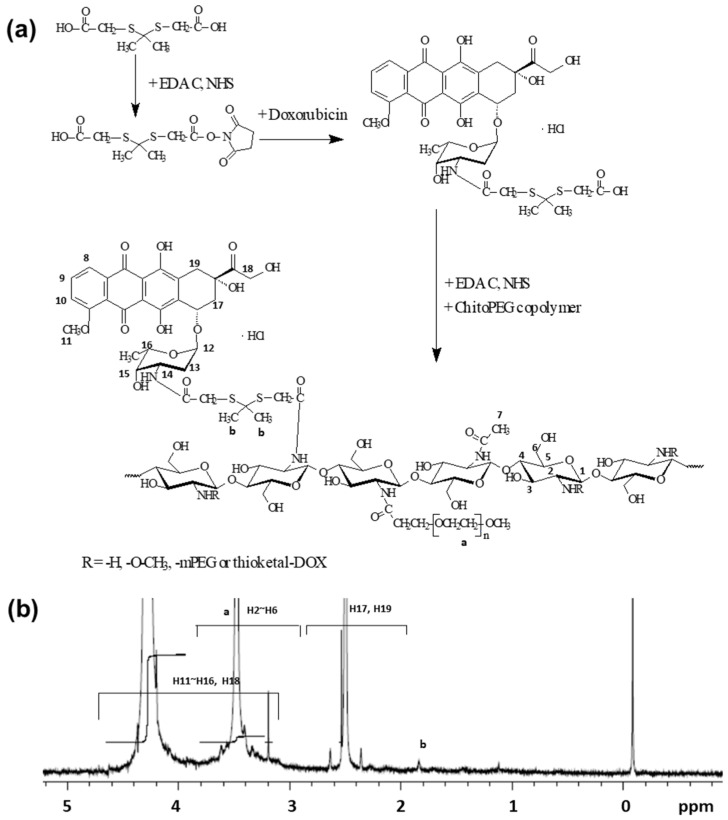
The synthesis scheme (**a**) and ^1^H NMR spectra (**b**) of the ChitoPEGthDOX conjugates.

**Figure 4 ijms-24-13704-f004:**
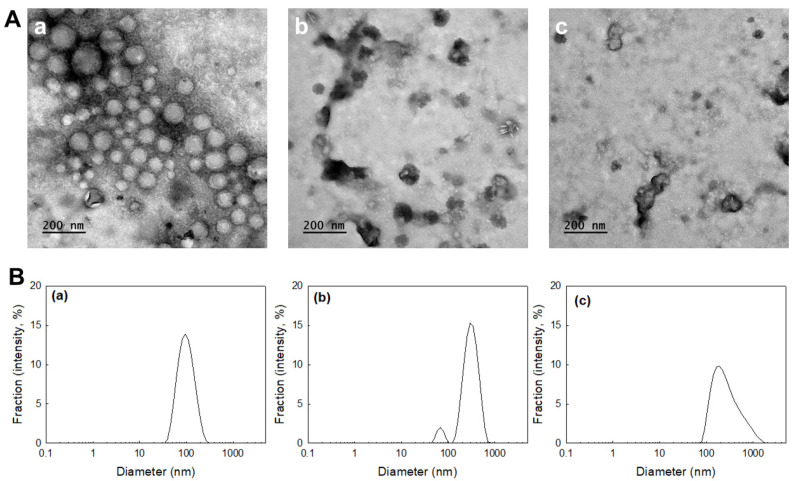
The effect of hydrogen peroxide on the physicochemical changes of the nanoparticles. (**A**) Morphological changes of the ChitoPEGthDOX nanoparticles. H_2_O_2_ concentration: (**a**) 0 mM; (**b**) 1.0 mM; (**c**) 10 mM. (**B**) Changes in the particle size of the ChitoPEGthDOX nanoparticles. H_2_O_2_ concentration: (**a**) 0 mM; (**b**) 1.0 mM; (**c**) 10 mM. ChitoPEGthDOX nanoparticles (1 mg/mL in PBS) were incubated with H_2_O_2_ for 3 h at 37 °C, and then the particle sizes and morphology were evaluated.

**Figure 5 ijms-24-13704-f005:**
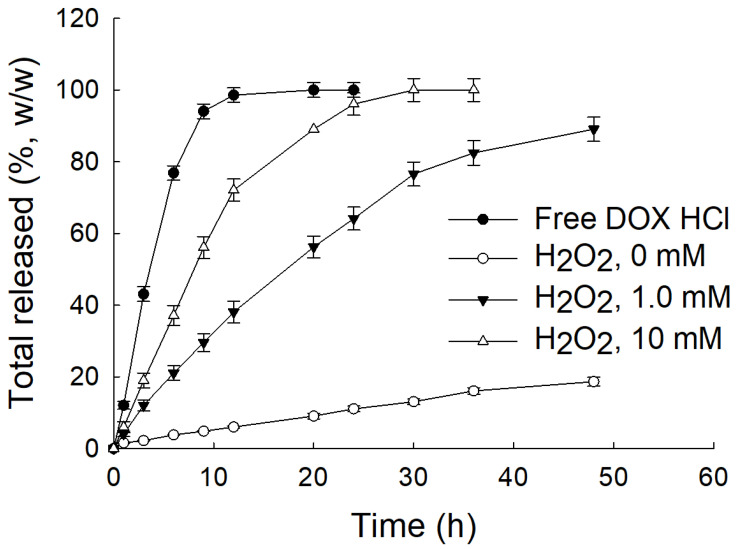
DOX release from the ChitoPEGthDOX nanoparticles.

**Figure 6 ijms-24-13704-f006:**
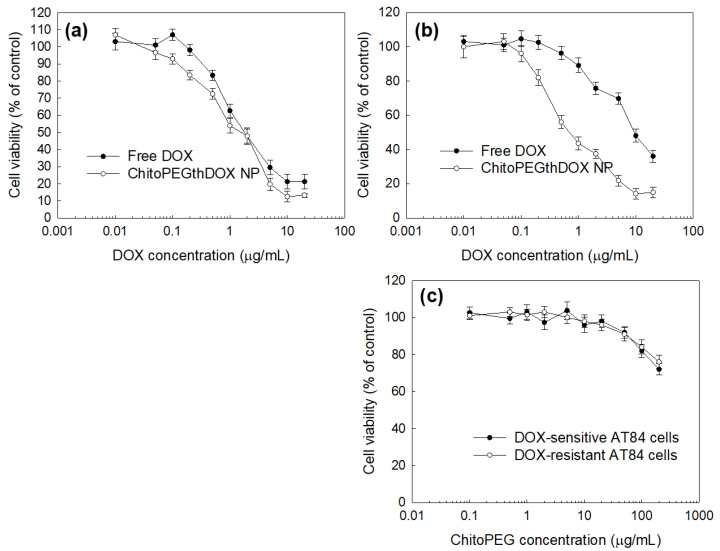
Anticancer activity of DOX and ChitoPEGthDOX nanoparticles (ChitoPEGthDOX NP) against DOX-sensitive AT84 cells (**a**) and DOX-resistant AT84 cells (**b**). The ChitoPEG copolymer, as an empty carrier, was treated to DOX-sensitive and DOX-resistant AT84 cells (**c**).

**Figure 7 ijms-24-13704-f007:**
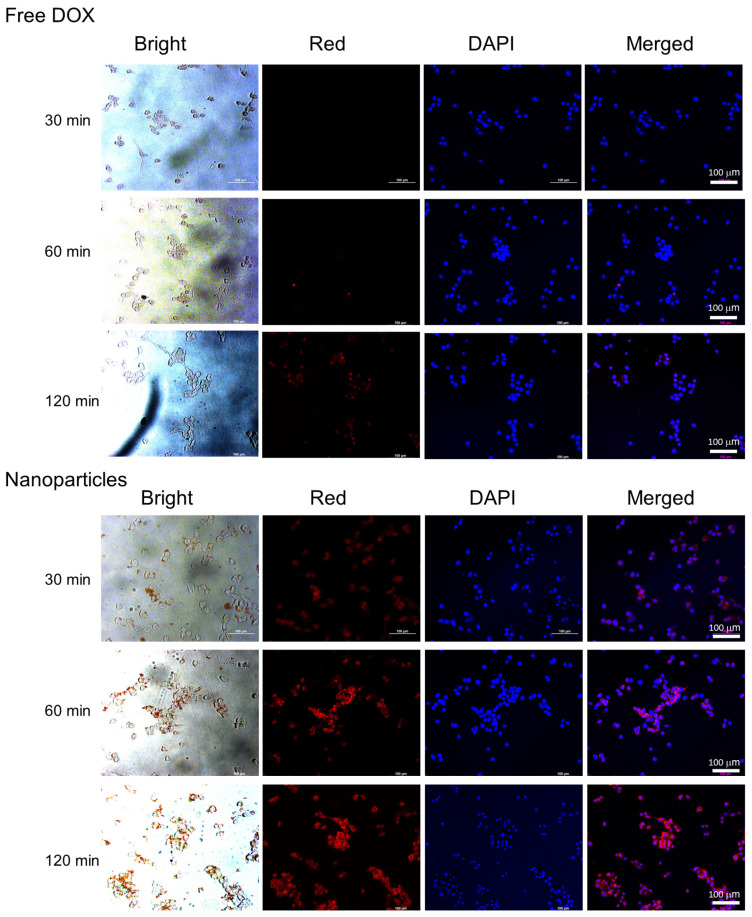
Fluorescence observation of DOX-resistant AT84 cells in vitro. DOX-resistant AT84 cells were exposed to DOX or nanoparticles for 0.5~2 h.

**Figure 8 ijms-24-13704-f008:**
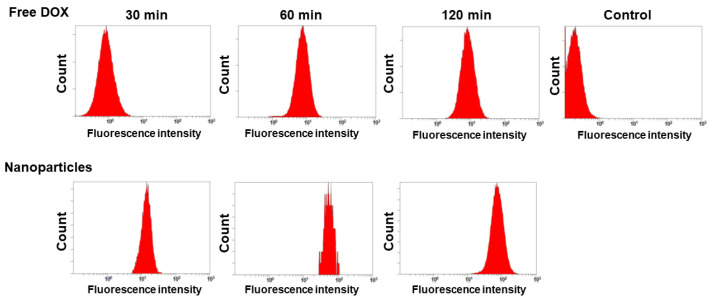
Flow cytometric analysis of free DOX and ChitoPEGthDOX nanoparticles.

**Figure 9 ijms-24-13704-f009:**
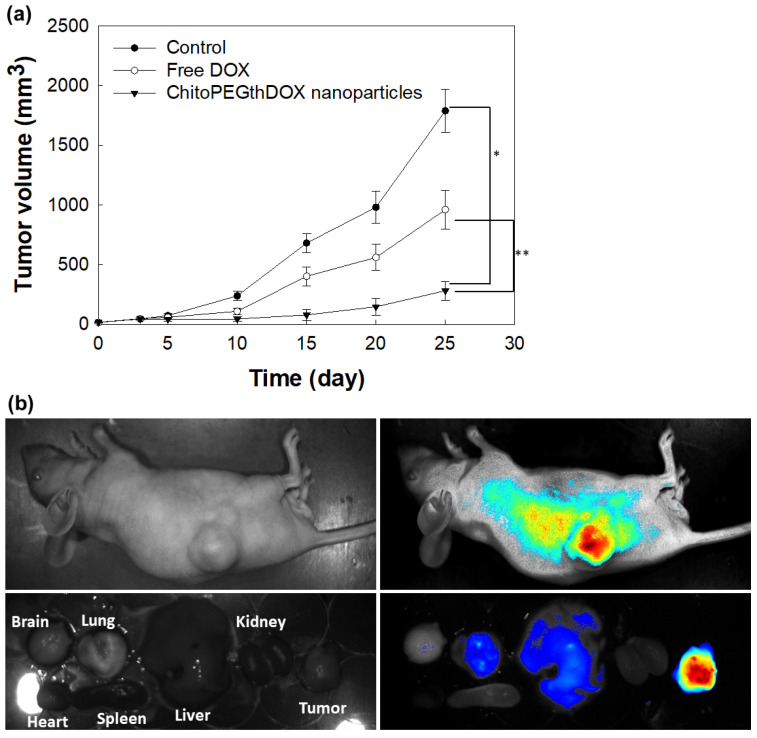
Anticancer activity of ChitoPEGthDOX nanoparticles against DOX-resistant AT84 tumor-bearing nude mice. (**a**) Tumor growth. ChitoPEGthDOX nanoparticles injected i.v. via the tail vein (injection volume: 100 µL; DOX dose, 5 mg/kg). Five mice were used for each treatment group and the result was expressed as average ± S.D. (**b**) Fluorescence imaging of the tumor-bearing mouse. The injection volume for both the DOX and nanoparticle solutions was 100 μL. ChitoPEGthDOX nanoparticles (5 mg/kg DOX concentration) were administered i.v. via the tail vein of the mouse. Then, 24 h later, the mouse was anesthetized with avertin for fluorescence imaging of the whole body and then sacrificed for the imaging of each organ. *, **; *p* < 0.01.

**Table 1 ijms-24-13704-t001:** Characterization of the ChitoPEGthDOX conjugate nanoparticles.

	Drug Contents (%, *w*/*w*)		Particle Size (nm) ^b^
	Theoretical ^a^	Experimental ^a^	
ChitoPEG copolymer		-	-
ChitoPEGthDOX nanoparticles	9.6	8.7	95.4 ± 2.1

^a^ Theoretical contents = [(Feeding DOX weight/(Feeding weight of ChitoPEG copolymer, ThdCOOH and DOX)] × 100. Experimental contents = [DOX weight/(ChitoPEGthDOX conjugates)] × 100. ^b^ Particle sizes were intensity fractions and the average ± S.D. from three measurements.

**Table 2 ijms-24-13704-t002:** IC_50_ values of the free DOX or ChitoPEGthDOX conjugates.

	IC_50_ (g/L) ^a^
DOX-sensitive cells Free DOX Nanoparticles ChitoPEG ^b^	1.9 1.7 -
DOX-resistant cells Free DOX Nanoparticles ChitoPEG copolymer	9.2 0.68 -

^a^ IC_50_ was derived from Figure 6. ^b^ Not determined. IC_50_ of the ChitoPEG copolymer as an empty carrier was higher than 100 g/L.

## Data Availability

Data sharing not applicable.

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
