# Peer review of "Redox-Sensitive Delivery of Doxorubicin from Nanoparticles of Poly(ethylene glycol)-Chitosan Copolymer for Treatment of Drug-Resistant Oral Cancer Cells"

_ijms, 2023, doi:10.3390/ijms241813704_

Round 1

Reviewer 1 Report

This report is interesting and to be use for the clinical application in future, I hope so.

Author Response

Review#1

This report is interesting and to be use for the clinical application in future, I hope so.

Answer) Thanks for your comment. According to your comment, we have plan to study pre-clinical evaluation of nanoparticles against small animal such as mouse, rat and/or rabbit. Furthermore, we revised the manuscript. Thanks again.

Reviewer 2 Report

In this manuscript, authors synthesized DOX-conjugated nanoparticles for reactive oxygen species (ROS) sensi- 12 tive delivery of doxorubicin (DOX) against DOX-resistant oral squamous carcinoma cells. For this 13 purpose, methoxy poly(ethylene glycol) (mPEG) was grafted to chitosan to produce chitosan-mPEG 14 (ChitoPEG) copolymer.
Based on these results, the authors suggest that ChitoPEGthDOX nanoparticles hold promise as a ROS-sensitive anticancer delivery system for DOX-resistant oral cancer cells.

- Comments to authors

1. What is the author trying to explain in line 41?

2. In line 325, there seems to be a typo in the unit representation for tumor volume (mm3).

3. What is the intended purpose of using bold text in Figure 7, table 1 and 2? Unnecessary bold text can confuse readers.

4. In Figure 6. footnote, the authors need to mention the graph (c).

The overall manuscript needs to be revised and reviewed so that readers can easily understand it.

Moderate editing of English language required

Author Response

Review#2

In this manuscript, authors synthesized DOX-conjugated nanoparticles for reactive oxygen species (ROS) sensi-12 tive delivery of doxorubicin (DOX) against DOX-resistant oral squamous carcinoma cells. For this 13 purpose, methoxy poly(ethylene glycol) (mPEG) was grafted to chitosan to produce chitosan-mPEG 14 (ChitoPEG) copolymer.
Based on these results, the authors suggest that ChitoPEGthDOX nanoparticles hold promise as a ROS-sensitive anticancer delivery system for DOX-resistant oral cancer cells.

- Comments to authors

  1. What is the author trying to explain in line 41?

Answer) Thanks for your comment. It is a typing error. I revised the manuscript according to your comment. Thanks again.

  1. In line 325, there seems to be a typo in the unit representation for tumor volume (mm3).

Answer) Thanks for your comment. It is a typing error. I revised the manuscript according to your comment. Thanks again.

The equation for the tumor volume: tumor volume (mm3) = (length × width2)/2.

  1. What is the intended purpose of using bold text in Figure 7, table 1 and 2? Unnecessary bold text can confuse readers.

Answer) Thanks for your comment. According to your comment, we revised the manuscript and bold text was changed. Thanks again.

Table 1. Characterization of nanoparticles of ChitoPEGthDOX conjugates.

Drug contents (%, w/w)

Particle size (nm) b

Theoretical a

Experimental a

ChitoPEG copolymer

9.6

-

8.7

-

ChitoPEGthDOX conjugates

95.4±2.1

a Theoretical contents = [(Feeding DOX weight/(Feeding weight of ChitoPEG copolymer, ThdCOOH and DOX)]* 100. Experimental contents = [Measured DOX weight/(ChitoPEGthDOX conjugates)]* 100.

b Particle sizes were intensity fractions and the average ± S.D. from three measurement.

Table 2. IC50 value of free DOX or ChitoPEGthDOX conjugates.

IC50 (g/L) a

DOX-sensitive cells

Free DOX

Nanoparticles

ChitoPEG b

1.9

1.7

-

DOX-resistant cells

Free DOX

Nanoparticles

9.2

0.68

-

a IC50 was derived from Figure 6.

b Not determined. IC50 of ChitoPEG copolymer as a empty carrier was higher than 100 g/L.

Figure 7. Fluorescence observation of DOX-resistant AT84 cells in vitro. DOX-resistant AT84 cells were exposed to DOX or nanoparticles for 0.5~2h.

  1. In Figure 6. footnote, the authors need to mention the graph (c).

Answer) Thanks for your comment.

Figure 6. Anticancer activity of DOX and ChitoPEGthDOX nanoparticles (ChitoPEGthDOX NP) against DOX-sensitive AT84 cells (a) and DOX-resistant AT84 cells (b). ChitoPEG copolymer as a empty carrier was treated to DOX-sensitive and resistant AT84 cells (c).

The overall manuscript needs to be revised and reviewed so that readers can easily understand it.

Answer) Thanks for your comment. According top your comment, we revised the manuscript in many paragraphs and, furthermore, grammatical error was also checked in all parts. Thanks for your comment again.

Reviewer 3 Report

The authors in this manuscript titled “ Redox-sensitive delivery of doxorubicin from nanoparticles of poly(ethylene glycol)-chitosan copolymer for treatment of drug-resistant oral cancer cells” have designed and validated a new nanoparticle based on Chitosan copolymer for the delivery of doxorubicin in oral cancer patients. The article described well the process of nanoparticle preparation and validated the activity in vitro in oral cancer cell line as well as in-vivo in a cell-derived xenograft model. The manuscript is interesting and should be considered for publication in IJMS after some major revisions. The authors should consider my specific comments for improving this manuscript.

1.       I don’t follow when authors don’t describe the figure legends well. It should be described in figures 1, 2, 3 & 4. What compound is presented and then a proper figure legend should be provided for better understanding.

2.       The authors should provide justification for using chitosan copolymer in this study.

3.       The authors mentioned once in line 117 that the particle size is less than 100 nm but then they mentioned later it is less than 200 nm (line 132). The authors should provide clarification for this discrepancy.

4.       In Figure 6, the authors have described in the text about using oral cancer cell line AT84, whereas in Figure two different cell lines are presented none of them is an oral cancer cell line. What is the reason for this discrepancy and why is it not mentioned in the text? Is it that this data is from some other experiment in a different study?

5.       The authors have not mentioned the sample size of their in-vivo study. They should clarify the sample size and then describe the significance of the treatment.

6.       There are a few English typo mistakes starting from line 41. The authors should re-read the manuscript and correct the typos for better clarification.

7.       The authors should pay attention that the abstract and conclusions are written exactly the same. The author should paraphrase or further improve the discussion in conclusions to ensure there is no overlap of sentences and repetition that looks like copy/paste.

Minor editing is required. There is a repetition of sentences that should be redefined in the conclusion section. 

Author Response

Review#3

The authors in this manuscript titled “Redox-sensitive delivery of doxorubicin from nanoparticles of poly(ethylene glycol)-chitosan copolymer for treatment of drug-resistant oral cancer cells” have designed and validated a new nanoparticle based on Chitosan copolymer for the delivery of doxorubicin in oral cancer patients. The article described well the process of nanoparticle preparation and validated the activity in vitro in oral cancer cell line as well as in-vivo in a cell-derived xenograft model. The manuscript is interesting and should be considered for publication in IJMS after some major revisions. The authors should consider my specific comments for improving this manuscript.

  1. I don’t follow when authors don’t describe the figure legends well. It should be described in figures 1, 2, 3 & 4. What compound is presented and then a proper figure legend should be provided for better understanding.

Answer) Thanks for your comment. According to your comment, we revised the manuscript and figure legends were checked/revised in the manuscript. Thanks again for your comment.

Figure 1. Synthesis scheme (a) and 1H NMR spectra (b) of ChitoPEG copolymer. N-hydroxysuccinimide (NHS)-activated mPEG (mPEG-NHS) was conjugated with amine groups of chitosan backbone to make ChitoPEG graft copolymer.

Figure 2. Chemical structure (a) and 1H NMR spectra (b) of ThdCOOH. Chemical structure (c) and 1H NMR spectra (d) of DOX HCl.

Figure 3. Synthesis scheme (a) and 1H NMR spectra (b) of ChitoPEGthDOX conjugates.

Table 1. Characterization of nanoparticles of ChitoPEGthDOX conjugates.

Drug contents (%, w/w)

Particle size

(nm) b

Theoretical a

Experimental a

ChitoPEG copolymer

-

-

ChitoPEGthDOX

conjugates

9.6

8.7

95.4±2.1

a Theoretical contents = [(Feeding DOX weight/(Feeding weight of ChitoPEG copolymer, ThdCOOH and DOX)]* 100. Experimental contents = [DOX weight/(ChitoPEGthDOX conjugates)]* 100.

b Particle sizes were intensity fractions and the average ± S.D. from three measurement.

  1. The authors should provide justification for using chitosan copolymer in this study.

Answer) Thanks for your comment. Chitosan copolymer means graft copolymer composed of chitosan and mPEG, i.e. mPEG was grafted to the backbone of chitosan. We indicated it in the manuscript (Results and discussion).

Nonionic and hydrophilic mPEG was introduced into the backbone of chitosan to synthesize graft copolymer composed of chitosan and mPEG (abbreviated as ChitoPEG copolymer) as shown in Figure 1(a).

  1. The authors mentioned once in line 117 that the particle size is less than 100 nm but then they mentioned later it is less than 200 nm (line 132). The authors should provide clarification for this discrepancy.

Answer) Thanks for your comment. According to your comment, we revised the manuscript. Practically, nanoparticles have small diameter less than 100 nm in average particle sizes and then phrase was changed to “dimater of nanoparticles size was around 100 nm”

Nanoparticles of ChitoPEGthDOX conjugates have small particle size around 100 nm as shown in Table 1.

Diameter of nanoparticles were around 100 nm.

  1. In Figure 6, the authors have described in the text about using oral cancer cell line AT84, whereas in Figure two different cell lines are presented none of them is an oral cancer cell line. What is the reason for this discrepancy and why is it not mentioned in the text? Is it that this data is from some other experiment in a different study?

Answer) Thanks for your comment. We apologize we did typing error. Then, we revised the manuscript and corrected them in the manuscript. Thanks for your comment again.

Figure 6. Anticancer activity of DOX and ChitoPEGthDOX nanoparticles (ChitoPEGthDOX NP) against DOX-sensitive AT84 cells (a) and DOX-resistant AT84 cells (b). ChitoPEG copolymer as an empty carrier was treated to DOX-sensitive and resistant AT84 cells (c).

  1. The authors have not mentioned the sample size of their in-vivo study. They should clarify the sample size and then describe the significance of the treatment.

Answer) Thanks for your comment. Sample sizes were same nanoparticles described in Figure 4 and Table 1. Particle size of nanoparticles were 95.4±2.1 nm. Furthermore, injection volume of solution was 100 microliter. Anyway, we described it in the manuscript and discussed more in Result and Discussion section.

The particle size is known to govern fate of nanoparticles in the biological system [29-31]. He et al. reported that particle size of nanoparticles govern in vivo fate after administration [30]. They argued that nanoparticles having 150 nm in particle size were efficiently accumulated in tumor tissue. Especially, Caster et al. reported the importance of particle size of nanoparticles in chemoradiotherapy, i.e. nanoparticles having 100 nm in diameter resulted in smallest tumor growth rate compared to nanoparticles having 50 nm or 150 nm in particle sizes [31]. Furthermore, they reported that smallest size of nanoparticles does not promise best anticancer activity even though they are effective to avoid hepatic and splenic accumulation. Our nanoparticles have small diameter about 100 nm in average particle sizes as shown in Figure 4 and Table 1. They have suitable particle sizes for site-specific targeting of tumor tissue and anticancer activity.

Figure 9. Anticancer activity of ChitoPEGthDOX nanoparticles against DOX-resistant AT84 tumor-bearing nude mice. (a) Tumor growth. ChitoPEGthDOX nanoparticles were i.v. injected via tail vein (injection volume: 100 µL; DOX dose, 5 mg/kg). Five mice were used for each treatment group and expressed as average ± S.D. (b) Fluorescence imaging of the tumor-bearing mouse. Injection volume of DOX solution and nanoparticle solution was 100 μL. ChitoPEGthDOX nanoparticles (5 mg/kg DOX concentration) were i.v. administered via the tail vein of the mouse. 24 h later, mouse was anesthetized with avertin for fluorescence imaging of whole body and then sacrificed for imaging of each organ. *, **; p < 0.01.

  1. Alexis F, Pridgen E, Molnar LK, Farokhzad OC. Factors affecting the clearance and biodistribution of polymeric nanoparticles. Mol Pharm. 2008 Jul-Aug;5(4):505-15.
  2. He C, Hu Y, Yin L, Tang C, Yin C. Effects of particle size and surface charge on cellular uptake and biodistribution of polymeric nanoparticles. Biomaterials. 2010 May;31(13):3657-66.
  3. Caster JM, Yu SK, Patel AN, Newman NJ, Lee ZJ, Warner SB, Wagner KT, Roche KC, Tian X, Min Y, Wang AZ. Effect of particle size on the biodistribution, toxicity, and efficacy of drug-loaded polymeric nanoparticles in chemoradiotherapy. Nanomedicine. 2017 Jul;13(5):1673-1683. 

  1. There are a few English typo mistakes starting from line 41. The authors should re-read the manuscript and correct the typos for better clarification.

Answer) Thanks for your comment. We apologize many typing errors. According to your comment, we checked typing errors and grammatical errors in whole manuscript and then expressed as red color. We revised the manuscript in all part. Thanks again for your comment.

  1. The authors should pay attention that the abstract and conclusions are written exactly the same. The author should paraphrase or further improve the discussion in conclusions to ensure there is no overlap of sentences and repetition that looks like copy/paste.

Answer) Thanks for your comment. According to your comment, we changed the abstract to distinguish with conclusion and for clear sentences. Thanks for your comment.

Abstract: Reactive oxygen species (ROS) sensitive polymer nanoparticles were synthesized for tumor targeting of anticancer drug, doxorubicin (DOX). For this purpose, DOX was conjugated to the backbone of chitosan/methoxy poly(ethylene glycol) (mPEG) (ChitoPEG) graft copolymer using thioketal linker. For ROS-sensitive delivery, thioketal linker was introduced between chitosan backbone and DOX. ChitoPEG graft copolymer was synthesized by conjugation of mPEG to the amine groups of chitosan backbone. DOX was conjugated with thioketal dicarboxylic acid (ThdCOOH) and then DOX-ThdCOOH conjugates were conjugated with ChitoPEG copolymer. After that, chemical structure of DOX-conjugated ChitoPEG copolymer (ChitoPEGthDOX) was confirmed with 1H nuclear magnetic resonance (NMR) spectra. Nanoparticles of ChitoPEGthDOX conjugates have spherical shapes and small sizes around 100 nm. In observation of transmission electron microscopy (TEM), ChitoPEGthDOX nanoparticles were disintegrated in the presence of hydrogen peroxide and particle size distribution was also changed from monomodal/narrow distribution pattern to multi modal/wide distribution pattern. Furthermore, DOX was released faster in the presence of hydrogen peroxide. These results indicated that ChitoPEGthDOX nanoparticles have ROS sensitivity. For anticancer activity evaluation of nanoparticles, AT84 oral squamous carcinoma cells were used and, furthermore, DOX-resistant AT84 cells were prepared in vitro. DOX itself and nanoparticles showed dose-dependent cytotoxicity both of DOX-sensitive and DOX-resistant AT84 cells in vitro. However, DOX itself showed reduced cytotoxicity against DOX-resistant AT84 cells while nanoparticles showed almost similar cytotoxicity both of DOX-sensitive and DOX-resistant AT84 cells. These results were due to that intracellular delivery of free DOX was inhibited while nanoparticles were efficiently internalized in DOX-resistant cells. In vivo study using DOX-resistant AT84 cell-bearing tumor xenograft model showed that nanoparticles have higher antitumor efficacy than those of free DOX treatment. These results were due to that nanoparticles were efficiently accumulated in the tumor tissue, i.e. fluorescence intensity in the tumor tissue was strongest than that of other organs. We suggest that ChitoPEGthDOX nanoparticles are promising candidate for ROS-sensitive anticancer delivery against DOX-resistant oral cancer cells.

Reviewer 4 Report

The introduction start with a part from another sentence. Seems that the first phrase is written only by reconstruction of phrase from another reference maybe the manuscript is not the last version before to be submitted.

Although the synthesis is well described , the method has no novelty.  The method can be applied for any other drug.

I consider the study is not characterized by  novelty, being more a routine one.

Author Response

Review#4

The introduction starts with a part from another sentence. Seems that the first phrase is written only by reconstruction of phrase from another reference maybe the manuscript is not the last version before to be submitted.

Answer) Thanks for your comment. According to your comment, we revised the manuscript and introduction part was fully rewritten to be clear the objectives of this study. Please consider our manuscript.

Although the synthesis is well described, the method has no novelty.  The method can be applied for any other drug.

Answer) Thanks for your comment. At this moment, we focused on the DOX delivery against drug-resistant oral cancer cells. Even though the method has little novelty, we approved that DOX-conjugated nanoparticles can be internalized into the drug-resistant cells with ROS-sensitive manner. Especially, release rate of DOX was controlled by hydrogen peroxide concentration. Anyway, we have plan to do another research for next paper using molecular targeted drug such as HDAC inhibitor and we will report them in the future. Thanks for your comment.

I consider the study is not characterized by novelty, being more a routine one.

Answer) Thanks for your comment. At this moment, we found that little references from “doxorubicin and thioketal group and chitosan” for ROS-sensitive delivery of anticancer drugs. In this study, we focused on the ROS-sensitive delivery of traditional anticancer drug, DOX against DOX-resistant oral cancer cells. We revised the manuscript fully. Thanks for your comment.

Round 2

Reviewer 2 Report

No issues with this revised manuscript.

Reviewer 3 Report

The authors have improved the manuscript very well and provided a clear justifications for all my previous comments.